# Self-Supervised Behavior Cloned Transformers are Path Crawlers for Text Games

**Ruoyao Wang** and **Peter Jansen**
University of Arizona, USA
{ruoyaowang,pajansen}@arizona.edu

## Abstract

In this work, we introduce a self-supervised behavior cloning transformer for text games, which are challenging benchmarks for multi-step reasoning in virtual environments. Traditionally, Behavior Cloning Transformers excel in such tasks but rely on supervised training data. Our approach auto-generates training data by exploring trajectories (defined by common macro-action sequences) that lead to reward within the games, while determining the generality and utility of these trajectories by rapidly training small models then evaluating their performance on unseen development games. Through empirical analysis, we show our method consistently uncovers generalizable training data, achieving about 90% performance of supervised systems across three benchmark text games.[1]

## 1 Introduction

Complex tasks often involve decomposing problems into sequential steps to accomplish a goal. This is particularly the case for text games, which are interactive virtual environments where an agent progressively selects actions until a task is complete. Examples of such tasks range from cooking a recipe (Côté et al., 2018), finding a treasure (Yuan et al., 2018), executing a science experiment (Tamari et al., 2021; Wang et al., 2022), to evaluating common sense reasoning (Shridhar et al., 2021; Murugesan et al., 2021; Gelhausen et al., 2022).

Text game agents are frequently modeled using reinforcement learning (e.g. He et al., 2016; Ammanabrolu and Hausknecht, 2020; Yao et al., 2020; Singh et al., 2021; Tuyls et al., 2022), wherein an agent explores an environment with the goal of learning a policy that chooses actions leading to reward and task completion. More recently, text game agents recast reinforcement learning as

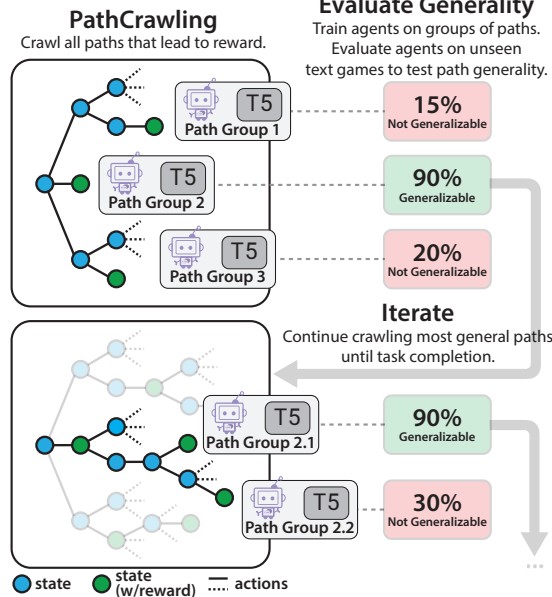

Figure 1: An overview of our approach. Initially, game trajectories from the current state are extracted up to a specified horizon, which extends to the first reward. The generality of these paths is assessed by training a compact T5 model and evaluating its performance on unseen development games. High-performing trajectories are subsequently extended through further exploration in an iterative process until the game is ultimately completed.

sequence-to-sequence problems using architectures like the Behavior Cloning Transformer (Torabi et al., 2018) to leverage the inference capability and robustness of modern transformers (Wang et al., 2023; Li et al., 2022; Lin et al., 2023). In this setup, the transformer is provided with an environment observation as input, and produces a string indicating the next action to take as output. A consequence of this reformulation is that, rather than exploring their environments, behavior cloning transformers require supervised training data, necessitating generating human gold playthroughs.

In Figure 1, we present our approach to overcome this data supervision limitation by automatically crawling, grouping, and evaluating candidate paths that can serve as useful and generalizable training data. By rapidly evaluating these candi-

---

[1] Released as open source: https://github.com/cognitiveailab/pathfinding-rl

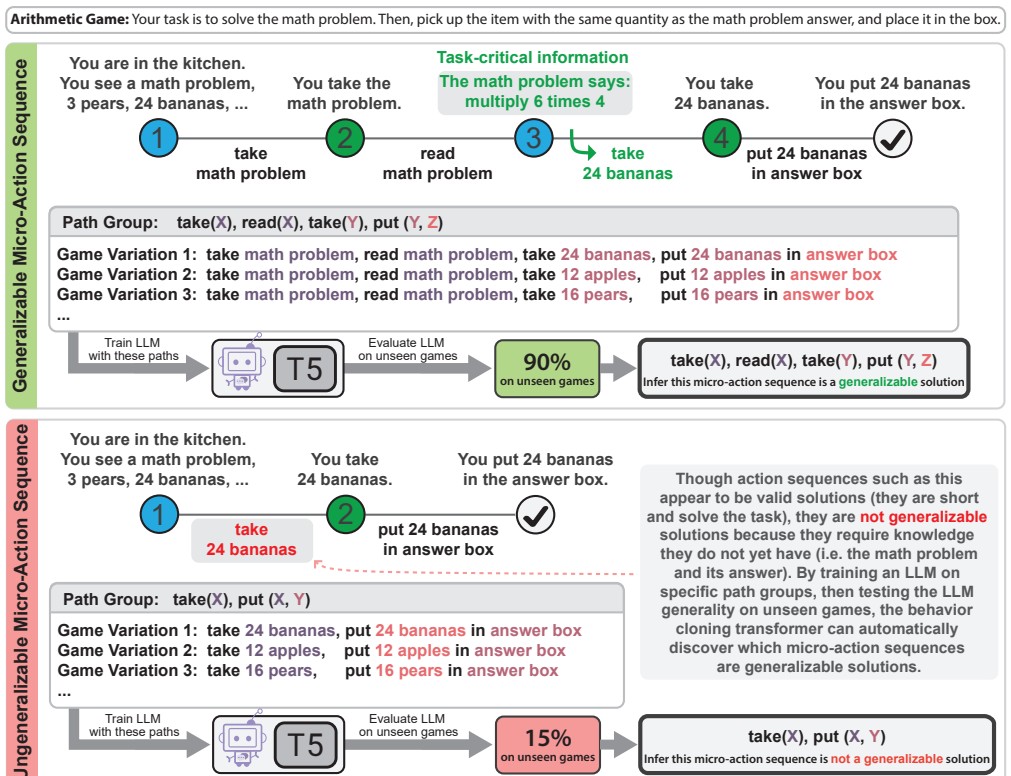

Figure 2: Two example path groups from the Arithmetic game, emphasizing that not all high-scoring paths serve as useful training data. Paths having the same macro-action sequence from parametric game variations are grouped together, as depicted above. These examples underscore the concept that shorter, quicker paths, like the one at the bottom, may lack generalizability, leading to poor model performance.

date sets of training data by training small models, we show how the performance of those models on unseen development games can be used as a self-supervision signal to help guide further path crawling, and discover useful training data automatically. This approach is validated on three benchmark games from the TEXTWORLDEXPRESS simulator (Jansen and Cote, 2023). Our contributions include: (1) demonstrating generating self-supervised training data through crawling, grouping, and evaluating game trajectories, (2) using the performance of small rapidly-trained models as self-supervision signals, and (3) developing heuristic methods to align and merge training data, reducing the task search space and ultimate training costs.

## 2 Approach

Figure 1 provides an overview of our approach. The goal of our work is to find a set of generalizable training data that can be a substitute for human gold playthroughs of text games, and suitable for training a behavior cloning transformer agent. While simply exhaustively pathcrawling all possible trajectories in a game offers a method to find winning paths, not all winning paths are gener-

alizable (see Figure 2), so such a method would yield poor training data, while also being computationally expensive. Here, we show that by training small models on candidate training data, and evaluating performance on unseen development games, we can quickly evaluate the generalizability of that data. In terms of tractability, by iteratively pathcrawling only up to the next reward, and then continuing pathcrawling only from the most generalizable paths, we can limit the size of the search to only paths most likely to be generalizable and form quality training data.

In this work, we use text games that have parametric variations of each task, such as different math problems to solve for an *arithmetic* game, which is standard practice for evaluating a model's generalization ability (e.g., Côté et al., 2018; Murugesan et al., 2021; Wang et al., 2022).

**Path Crawling:** Our process begins with a path crawler exploring all potential trajectories in a game from the start state until achieving non-zero reward or reaching a limited horizon. This crawler also navigates multiple parametric game variations, each with distinct objectives, such as *making a stirfry* versus *baking a cake* in a cooking game.

**Path Grouping into Parameterized Macro-Action Sequences:** Text game actions consist of a command verb (e.g., *take, put, open*) with one (e.g. *open closet*) or two (e.g. *put apple in cupboard*) arguments, where a sequence of actions (i.e. a full or partial playthrough) is called a *trajectory*. When creating candidate training sets for the behavior cloning transformer, we attempt to group together trajectories that – as best as we can tell – appear to solve different parametric variations of a game using roughly the same sequence of actions. We do this by abstracting the trajectories into variabilized action sequences. For example, in the TEXTWORLD COMMON SENSE game, a trajectory such as *"take hat", "put hat on hat rack"* from one variation, and *"take dirty shirt", "put dirty shirt in washer"* from another variation, could both be abstracted into the same *parameterized macro-action sequence* "TAKE(X), PUT(X, Y)". We call a set of trajectories from different game variations that can all be described using the same parameterized macro-action sequence a *path group*. In the experiments reported below, path groups are typically created by grouping paths from 100 training variations of a game.

**Evaluating Path Groups:** The above process yields $N$ path groups, each representing a unique parameterized macro-action sequence. These groups provide the training data for a behavior cloning model. Although $N$ is typically large, we've observed that generalizable sequences are often among the shorter paths (though not the shortest – Figure 2 provides a counterexample). For efficiency, we select the $K$ shortest path groups and train a T5-based behavior cloning transformer individually on each, assessing the model performance on 100 unseen game variations from the development set.[2] The process continues until we identify a group surpassing a predefined performance threshold $T$. If no single group meets this early stopping criterion, we attempt to merge path groups to discover higher-scoring combinations, detailed further in APPENDIX C.2.1.

**Incremental Path Crawling:** Reinforcement learning problems can offer sparse or dense rewards. To minimize the search space and the total number of path groups to evaluate, we initially crawl and assess paths up to the first reward instance. Upon identifying a path group with gener-

---

[2] An analysis of performance versus number of parametric variations included in training is provided in the APPENDIX.

alizable performance to this point, the model continues path crawling, commencing with this macro-action sequence and assessing generality until the next reward signal. This cycle persists until reaching a winning state, effectively segmenting the task into subtasks demarcated by non-zero rewards.

**Benchmarks:** Our exploration of using path crawling for generating self-supervised training data spans three benchmark text games. TEXTWORLD COMMON SENSE (TWC) (Murugesan et al., 2021) tasks agents with assigning household items (e.g. *a dirty shirt*) to their canonical locations (e.g. *a washing machine*). ARITHMETIC engages agents in a math problem, followed by a pick-and-place task using the calculated result. SORTING requires agents to arrange objects by quantity. Each game presents parametric variations altering the environment and task-specific objects across episodes. For all experiments, we generate distinct training, development, and test sets, each with 100 variations of each game. Objects crucial to tasks are unique across sets. Refer to the APPENDIX for game specifics and playthrough examples.

## 3 Models

We assess four models to showcase the effectiveness of the self-supervised behavior cloning transformer. Further model details and hyperparameters can be found in the APPENDIX.

**Supervised Behavior Cloning Transformer:** This supervised baseline employs a sequence-to-sequence model that packs the game task description, current and previous observations, and prior action into a prompt, while training it to generate the agent's next action. Gold paths for training are generated by gold agents. At inference time candidate next-actions are generated using beam search, and the first valid action is used. For efficiency, all reported experiments employ the T5-BASE (220M parameter) model (Raffel et al., 2020).

**Self-supervised Behavior Cloning Transformer:** This model, the focus of our work, parallels the supervised transformer, with the exception being its training data is generated via the path crawling method described in Section 2.

**Additional comparisons:** Although our aim is to showcase behavior cloning self-supervision, we offer additional baselines to contextualize the results. The DRRN (He et al., 2016) is a strong reinforcement learning baseline that separately encodes

| Benchmark | DRRN Baseline | | GPT-4 Baseline | | Behavior Cloned Transformer | | | |
|---|---|---|---|---|---|---|---|---|
| | | | | | Supervised | | Self-Supervised | |
| | Score | Steps | Score | Steps | Score | Steps | Score | Steps |
| Arithmetic | 0.17 | 10 | 0.76 | 15 | 0.54 | 5 | 0.44 | 5 |
| Sorting | 0.03 | 21 | 0.04 | 7 | 0.72 | 7 | 0.59 | 6 |
| TWC | 0.57 | 27 | 0.90 | 6 | 0.90 | 6 | 0.89 | 15 |
| Average | 0.26 | 19 | 0.57 | 9 | 0.72 | 6 | 0.64 | 9 |

Table 1: Model performance across three text game benchmarks, assessed on 100 unseen parametric test set variations. The score (0-1) represents game progress, while steps denote the average game steps taken to reach a final state—lower is better. The self-supervised transformer closely mirrors the supervised model's performance, marginally surpasses the GPT-4 baseline, and significantly outperforms the DRRN reinforcement learning baseline.

observations and actions into different embedding spaces, then learns a policy that selects actions at each step that maximize reward. Despite its age, the DRRN maintains near state-of-the-art performance across many text game benchmarks (e.g. Xu et al., 2020; Yao et al., 2020; Wang et al., 2022) – see Jansen (2022) for a review. For further context, we include a zero-shot GPT-4 baseline (OpenAI, 2023), a state-of-the-art model with an undisclosed parameter size and training data. More details are provided in the APPENDIX.

## 4 Results and Discussion

Table 1 shows the overall performance for all models. The DRRN reinforcement learning baseline yields modest overall performance, scoring an average of 0.26 across all benchmark text games and taking an average of 19 steps per game episode before completion. The supervised behavior cloning transformer significantly outperforms this, achieving a score of 0.72 across all games, effectively tripling task performance while also reducing the average steps required per episode to 6 – though at the expense of requiring gold training data. The GPT-4 baseline underperforms at the sorting game but excels in the other games, resulting in an average score of 0.57 and requiring 9 steps to finish. The self-supervised behavior cloning transformer achieves a score of 0.65, equivalent to 90% of the supervised system's performance, but uses entirely self-generated training data. It substantially outperforms the reinforcement learning baseline (which also lacks supervision), improving performance by a factor of 2.5 while generating solutions that are twice as efficient. The self-supervised behavior cloning transformer meets or exceeds the GPT-4 baseline on two of three games, despite the significant difference in model size, while only underperforming on the benchmark requiring mathematical

reasoning – for which smaller models tend to perform poorly (Razeghi et al., 2022).

**How do the trajectories identified by self-supervision compare to gold trajectories?** As shown in Table 2, self-supervised paths closely match gold paths, although they occasionally include task-irrelevant actions (e.g. *look around*) which diminish their efficiency. Given that the behavior cloning method models games as a Markov decision process (limited by input length to depend only on the previous state), these irrelevant actions may hinder the agent's recall of vital action history information, contributing to the slight performance drop compared to the supervised model.

**How can the performance of this method versus the DRRN and GPT-4 be contextualized?** Our goal in this work is to demonstrate self-supervision of a behavior cloning transformer using the pathcrawling method described in Section 2. Nevertheless, other baselines offer valuable context. The poor performance of the DRRN, a robust reinforcement learning baseline for text games, underscores the difficulty of exploring expansive search spaces without guiding context – one of the motivating factors towards the application of pretrained language models to guide exploration more effectively. On the other hand, while the GPT-4 model performs comparably to the self-supervised behavior cloning model, it is likely to be at least three orders of magnitude larger. This highlights the potential for small models to perform well given suitable exploration mechanisms.

**How many training examples are required to learn a task?** Figure 3 shows the performance of the T5-agent trained with varying numbers of gold trajectories. Generally, task performance increases roughly linearly with the number of training examples provided. An exception to this is the ARITHMETIC game, where nearly all 100 training

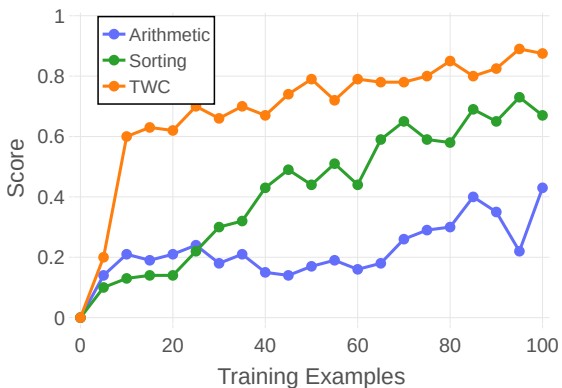

Figure 3: Performance of the supervised behavior cloning agent on each of the three benchmark games, when trained with varying numbers of training examples. Performance is reported on the development set.

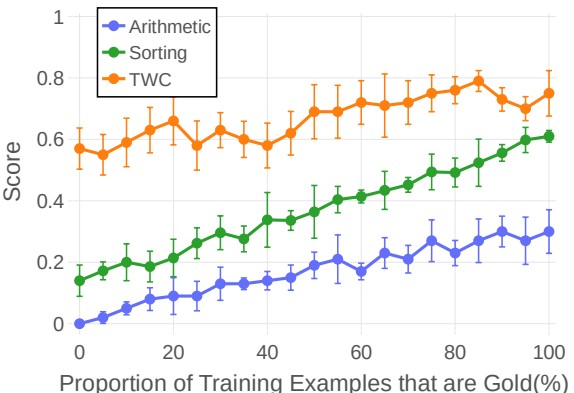

Figure 4: Performance of the supervised behavior cloning agent on each of the three benchmark games, when trained with varying proportion of gold trajectories interspersed with randomly sampled non-gold trajectories. Error bars represent the standard deviation of 5 runs per data point on the development set.

| Arithmetic | |
|---|---|
| SS | TAKE(X), READ(X), LOOK-AROUND, TAKE(Y), PUT(Y, Z) |
| Gold | TAKE(X), READ(X), TAKE(Y), PUT(Y, Z) |
| **Sorting** | |
| SS | TAKE(X), PUT(X, Y), TAKE(Z), PUT(Z, Y), TAKE(A), ... |
| Gold | TAKE(X), PUT(X, Y), TAKE(Z), PUT(Z, Y), TAKE(A), ... |
| **Text World Common Sense** | |
| SS | (TAKE(X), LOOK-AROUND, PUT(X, Y)) |
| | (TAKE(X), INVENTORY, OPEN(Y), PUT(X, Y)) |
| Gold | (TAKE(X), PUT(X, Y)) |
| | (TAKE(X), OPEN(Y), PUT(X, Y)) |

Table 2: Paths identified by our self-supervised method *(SS)* compared to gold paths for each game, showing that self-supervised paths are nearly identical to gold paths. Paths in parenthesis represent merged path groups.

**ries?** How easily can we find generalizable training data for a given game, amongst all its winning trajectories? To investigate this, we trained models with varying amounts of gold (generalizable) trajectories, versus other randomly sampled winning (but typically not generalizable) trajectories. At one extreme, the model is trained using entirely randomly sampled winning trajectories, while at the other extreme, the model is trained entirely using gold trajectories. The results, shown in Figure 4, show that the self-supervision method provides a large benefit, increasing performance between 15% and 43% across all games – highlighting the benefit of our self-supervision method.

**How is this method affected by training hyperparameters?** We have observed substantial differences in performance across various training hyperparameters (such as number of training epochs, or training random seed), and suspect part of the difference between supervised and self-supervised performance may be due to these fluctuations.

## 5 Conclusion

We present a self-supervised behavior cloning transformer for text games, that leverages a process of incremental path crawling in action spaces leading to rewards, while evaluating the generality of groups of crawled paths on unseen games, to generate high-quality training data. Our method achieves comparable performance to models trained on gold trajectories, efficiently exploring the environment without needing human playthroughs or gold agents. Our model excels over a robust reinforcement learning baseline and attains about 90% of supervised models' performance, emphasizing the potential of efficient exploration mechanisms, even for smaller models.

examples are needed to gain moderate performance. We hypothesize that this is a result of the pretrained model generally performing poorly at mathematical tasks. These results suggest that, pragmatically, using the self-supervised behavior cloning transformer may be possible with less training data on some environments, while other environments (such as ARITHMETIC) may benefit from more training examples. When using this self-supervised system in practice (i.e. on environments without gold training data), the end user may wish to tune the number of training examples they use as a hyperparameter in their model – though generally, the larger the number available, the more accurately the path crawler will be able to estimate performance.

**How effective is the data grouping method compared to randomly sampled winning trajecto-**

## Limitations

We present a method for generating self-supervised training data for behavior cloning transformer models, presented in the context of text games. Behavior cloning transformers (Torabi et al., 2018) and other frameworks for modeling reinforcement learning problems with transformers (including Decision Transformers (Chen et al., 2021) and Trajectory Transformers (Janner et al., 2021)) can outperform other reinforcement learning models in specific contexts, but still face a number of central challenges and limitations that depend upon the complexity of the problem space. The self-supervised behavior cloning transformer we present in this work has similar limitations, as well as other limitations that result from the computational complexity of path crawling.

**Large Action Spaces:** At each time step, the environment simulator provides a set of valid actions that are possible to take. In text games, this *action space* frequently contains ten or more actions, each of which typically takes one or two arguments that represent objects in the environment – while extreme cases can contain 70 or more actions, and up to 300 action templates (Hausknecht et al., 2020). As such, action spaces can quickly become large, containing thousands (or hundreds of thousands) of possible valid action-object combinations per step. While reinforcement models typically struggle with exploring these large action spaces resulting in low model performance, the path crawling model presented in this work would struggle to crawl very large action spaces, and may fail to find any winning paths within a given time budget. In practice, with the current model, we have found that action spaces containing approximately 500 valid actions per step are intractable to crawl beyond 3 steps (i.e. above $10^8$ states per episode).

**Sparse Rewards:** Tasks with sparse rewards are frequently challenging for reinforcement learning models. This work separates path crawling and next-action generation, making problems with moderately sparse rewards (i.e. within 4-5 steps) generally quick to find. However, for tasks with large action spaces and extremely sparse rewards, the path crawling procedure would become intractable.

**Complex, Varied Trajectories:** To evaluate path generality, this work groups paths by abstracted parameterized macro-action sequences. In the three benchmark tasks investigated in this work, the tasks can be solved by merging a small set of groups (for example, for TEXTWORLD COMMON SENSE, two groups are required: (1) TAKE(X) OPEN(Y) PUT(X,Y), which picks up an object, opens its container, then places it in the container, and (2) TAKE(X) PUT(X, Y), for cases where a container doesn't need to be opened, like a table). For more complex tasks that require a large number of different actions to solve across game episodes, this technique would require increasing the amount of training data provided, proportional to the number of groups estimated to be required to solve the task.

## Ethics Statement

**Broader Impacts:** Large language models typically perform complex multi-step inference in a manner that isn't easily possible to inspect or evaluate. Framing tasks as text games in embodied virtual environments helps make the inference steps explicit, such that the choices a model makes at each step are auditable and interpretable. This helps expose potential issues in reasoning in models. For example, language models can correctly answer more than 90% of multiple choice science exam questions, suggesting they can perform complex reasoning, but fail to answer most of those same questions when they are reframed as text games that require explicit multi-step reasoning (Wang et al., 2022).

**Intended Use:** This work presents a self-supervised system that generates its own training data based on paths that a language model finds generalizable. If applied more broadly to general reinforcement learning problems apart from text games, the model has the potential to find shortcuts through the action space that lead to task solutions based on a language model's existing competencies, but to do so through biased or harmful means. As such, for tasks that have the possibility of producing harm to specific groups, the training data produced by self-supervision should be manually inspected by a human to evaluate for bias or other potential harms. Because this work groups solutions into abstracted sequences of actions (i.e. path groups with similar solution methods, instantiated with different specific arguments, it may offer a means of inexpensively manually evaluating self-supervised training data – helping speed the identification of bias by streamlining the review of self-supervised training data for tasks modelled using self-supervised behavior cloning models.

## Acknowledgements

Thanks to Marc-Alexandre Côté, Raj Ammanabrolu, and the anonymous reviewers for helpful comments. Thanks to Dan Yang for drawing the little robot in Figure 1. This work supported in part by National Science Foundation (NSF) award #1815948 to PJ, and gift from the Allen Institute for Artificial Intelligence (AI2).

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

## A  Additional Experiment Information

The heuristic method we employ investigates shorter paths (i.e. macro-action sequences that take fewer actions to reach reward states) first. After grouping paths, the number of initial groups to explore for each game is 112 for TWC, 220 for ARITHMETIC, and 7 for SORTING.

---

**Text World Common Sense (TWC) Game**

**Task Description:** Your task is to pick up objects, then place them in their usual locations in the environment.

You are in the bedroom. In one part of the room you see **a wardrobe** that is closed. There is also a desk chair, that has nothing on it. You also see a dressing table, that has nothing on it. In another part of the room you see a chest of drawers that is closed. In one part of the room you see a bed, that has nothing on it. There is also a desk, that has nothing on it. You also see a night stand, that has nothing on it. In another part of the room you see **a clean brown shirt**.
Inventory:
  Your inventory is currently empty.
> *take clean brown shirt*

You take the **clean brown shirt**.
Inventory:
  **a clean brown shirt**
> *open wardrobe*

You open the wardrobe. It's empty inside.
Inventory:
  **a clean brown shirt**
> *put **clean brown** in **wardrobe***

*Game completed.*

---

Table 3: An example of a TEXTWORLD COMMON SENSE (TWC) game. *Actions* for the solution are italicized. Relevant **task objects** and **target locations** are bolded, while distractor locations are underlined.

## B  Environments

### B.1  Text World Common Sense (TWC)

TEXTWORLD COMMON SENSE (Murugesan et al., 2021) is a common sense reasoning benchmark in which the agent needs to collect items in a household environment and put them in their cannonical ("common-sense") locations. In the experiments reported in this paper, we generate parametric episodes containing a single room with one target item in it, roughly corresponding to the "easy" setting in Murugesan et al. (2021). The agent receives a score of 0.5 for picking up the target item, then an additional 0.5 score for putting that object in the correct location. An example playthrough of TEXTWORLD COMMON SENSE is shown in Table 3.

### B.2  Arithmetic

The ARITHMETIC game requires the agent to read then answer an arithmetic question (e.g. *add 42 and 9*), then put an item from the environment whose quantity matches the answer (e.g. *51 pineapples*) into a container called the answer box. Each episode contains a number of distractor items whose quantity corresponds to performing the arithmetic incorrectly (e.g. *33 peaches*, corresponds to subtracting 9 from 42). The agent receives a score of 0.5 for first reading the math problem, and will

| **Arithmetic Game** |
| --- |
| **Task Description:** Your first task is to solve the math problem. Then, pick up the item with the same quantity as the math problem answer, and place it in the box. |
| You are in the laundry room. In one part of the room you see a bench that has 378 apples on it. There is also a laundry basket that has **51 pineapples**, 4 bananas, and 16 oranges on it. You also see a work table that has 33 peaches, and 26 strawberries on it. In another part of the room you see a washing machine that is closed. In one part of the room you see a box, that is empty. There is also a math problem. You also see a clothes drier that is closed.
Inventory:
  Your inventory is currently empty.
> *take math problem* |
| You take the math problem.
Inventory:
  a math problem
> *read math problem* |
| Your task is to solve the following math problem: add 42 and 9 . Then, pick up the item with the same quantity as the answer, and place it in the box.
Inventory:
  a math problem
> *take 51 pineapples* |
| You take the **51 pineapples**.
Inventory:
  a math problem
  **51 pineapples**
> *put 51 pineapples in box* |
| *Game completed.* |

Table 4: An example ARITHMETIC game. *Actions* for the solution are italicized. Relevant **task objects** are bolded, while distractor objects are underlined.

receive a full score of 1.0 for placing the correct item into the answer box, regardless of whether it first reads the math problem or not. Placing incorrect items into the answer box will immediately result in task failure. An example playthrough of ARITHMETIC is shown in Table 4.

## B.3 Sorting

In the SORTING game, the agent needs to place items of different quantities into a special container (the answer box) in ascending order. For most parametric episodes, the item quantities include the additional challenge of containing different units (e.g. *23mg of oak* vs *2g of marble*) that the agent needs to interpret and normalize to correctly solve the task. Episodes randomly contain between 3 and 5 items that need to be sorted. The agent receives a partial reward ($1/\#items$) each time it places the correct object (i.e. next-largest quantity) into the answer box. The task fails any time it places items in the answer box in an incorrect order. An example playthrough of SORTING is shown in Table 5.

| **Sorting Game** |
| --- |
| **Task Description:** Your task is to sort objects by quantity. First, place the object with the smallest quantity in the box. Then, place the objects with the next smallest quantity in the box, and repeat until all objects have been placed in the box. |
| You are in the living room. In one part of the room you see a TV stand that has **23mg of oak** on it. There is also a coffee table, that has nothing on it. You also see a book case, that has nothing on it. In another part of the room you see a box, that is empty. In one part of the room you see a sofa that has **2g of marble** on it. There is also a end table that has **38mg of marble** on it. You also see a arm chair that has **39mg of wood** on it. In another part of the room you see a wastepaper basket that is closed. In one part of the room you see a side table that has **9g of oak** on it.
Inventory:
  Your inventory is currently empty.
> *take 23mg of oak* |
| You take the **23mg of oak**.
Inventory:
  **23mg of oak**
> *put 23mg of oak in box* |
| You put the **23mg of oak** in the box.
Inventory:
  Your inventory is currently empty.
> *take 38mg of marble* |
| You take the **38mg of marble**.
Inventory:
  **38mg of marble**
> *put 38mg of marble in box* |
| You put the **38mg of marble** in the box.
Inventory:
  Your inventory is currently empty.
> *take 39mg of wood* |
| You take the **39mg of wood**.
Inventory:
  **39mg of wood**
> *put 39mg of wood in box* |
| You put the **39mg of wood** in the box.
Inventory:
  Your inventory is currently empty.
> *take 2g of marble* |
| You take the **2g of marble**.
Inventory:
  **2g of marble**
> *put 2g of marble in box* |
| *Game completed.* |

Table 5: An example SORTING game. *Actions* for the solution are italicized. Relevant **task objects** are bolded.

## C Baselines

### C.1 Supervised Behavior Cloning Transformer

The input (prompt) of the T5 model is formatted as:

$d$ **** **OBS** $o_t$ **** **INV** $o_t^{inv}$ **** **LOOK** $o_t^{look}$ **** **<extra_id_0>** **** **PACT** $a_{t-1}$ **** **POBS** $o_{t-1}$ ****

where **** is the special token for separator and **<extra_id_0>** is the special token for the mask of text to generate used by the T5 model. $d$ is the task description. $o_t$, $o_t^{inv}$, $o_t^{look}$, $a_{t-1}$, and $o_{t-1}$ represent the current observation, current inventory information, current room description obtained by

the action "look around", last action, and last observation respectively. The separators **OBS**, **INV**, **LOOK**, **PACT**, and **POBS** are their corresponding special tokens.

During development, we train the behavior cloning T5-BASE model from 2 to 20 epochs, then choose the model that achieves the highest development score for testing. The models are tested on 100 unseen test variations to get the final performance. All three game environments have a step limit of 50 steps, which means if an agent does not complete within 50 steps, the score at the last step will be considered as the final score and the environment will be reset.

The T5 model is trained on gold paths generated by gold agents provided by the TEXTWORLDEX-PRESS simulator. At inference time, the a prompt is encoded from the observation of the current game state, and possible candidate next-actions are generated using beam search. We use a beam size of 8 and select the first valid action. If no valid actions are present, the model picks the valid action from the game environment with the highest non-zero unigram overlap to the candidate actions generated during beam search.

## C.2 Self-supervised Behavior Cloning Transformer

During the group evaluation stage, we take the top 10 shortest paths to evaluate for all three games. To vastly reduce computation time, the behavior cloning T5-models are not tuned but rather trained statically for 20 epochs for each path group. Performance is evaluated on 100 unseen development variations. The input strings are formatted the same as the supervised behavior cloning transformer. A group that achieves a development score higher than a threshold $T$ will be accepted. In our experiments, we use $T = 0.95$ for the TEXTWORLD COMMONSENSE, $T = 0.4$ for the ARITHMETIC game, and $T = 0.5$ for the SORTING game. In practice, one cycle of training a behavior cloning agent on a candidate set of paths then evaluating on 100 unseen development variations takes approximately 12 minutes on desktop hardware (i.e. RTX 4090), allowing experiments to complete in approximately 2 to 4 hours.

### C.2.1 Merging Multiple Solution Paths

Frequently, games may not be solvable with paths from a single group. For example, in TEXTWORLD COMMON SENSE, some items require that their container be opened before they can be placed inside (i.e. TAKE(X) OPEN(Y) PUT(X, Y)), while others can be directly placed in their winning locations (e.g. TAKE(X) PUT(X, Y)). Where a single path group does not solve all development variations, we take training data from the highest-performing variations of multiple path groups, and use these to assemble a final higher-performing training set.

We take the top 5 performing groups based on their development scores and combine each two of them to get 10 merged groups (i.e. $\binom{5}{2}$). Trivially combining the training data from multiple groups does not work well, because each of these groups may contain both generalizable paths, as well as paths that fail to generalize on episodes where another group of data is effective. Instead, we combine groups by choosing individual trajectories from each group that perform well on their respective development episodes. If two groups have paths that perform well for a given development episode, we choose the trajectory that has the highest development score. While this method has the benefit of being pragmatic (i.e. it chooses to combine paths that perform well empirically), it presents the challenge of potentially overfitting on the unseen development set, which may limit ultimate generality when the model is eventually evaluated on the test set. To mitigate this, when groups need to be merged, we flip the training and development sets, so that the path crawler is always generating hypotheses on one set, and evaluating them on a different, unseen set. We ultimately choose the merged group with the highest score on the development set as the final training set, and use this to generate final performance figures on the unseen training set.

## C.3 Deep Reinforcement Relevance Network (DRRN)

We make use of an existing benchmark implementation of the DRRN model[3]. At each time step, the task description, observation, agent inventory information, and room description are appended into a single string (as in the case of the behavior cloning models) and encoded by a GRU. Valid actions at each step are encoded by a separate GRU. A Q-network, which consists of two linear layers, estimates the Q-value of each action using the encoded observation-action pair. The action with the

---

[3] https://github.com/microsoft/tdqn

highest estimated Q-value will be chosen as the next action. We trained the DRRN for 100K steps. 16 environments are trained in parallel. The final results represent the average performance of 5 different models trained using different starting random seeds.

## C.4 GPT-4

The prompt of the GPT-4 agent includes the all the information in the behavior cloning transformer's prompt, which are the task description, the current observation, current inventory information, current room description obtained by the *"look around"* action, the last action, and the last observation. We also offer a full list of valid actions at each step and ask GPT-4 to choose one action from the list. An example of the prompt used for the GPT-4 agent is found below:

---

**GPT-4 Agent Prompt**

You are playing a text game. Choose the proper action from the list of valid actions to win the game.

Task description:
Your first task is to solve the math problem. Then, pick up the item with the same quantity as the math problem answer, and place it in the box.

Observation:
You take the math problem.
You are in the supermarket. In one part of the room you see a box, that is empty. There is also a showcase that has 29 blueberries, 42 avocados, 22 cabbages, 88 eggplants, 46 peas, and 17 peppers on it.

Inventory:
    a math problem

Previous observation:
You are in the supermarket. In one part of the room you see a box, that is empty. There is also a math problem. You also see a showcase that has 29 blueberries, 42 avocados, 22 cabbages, 88 eggplants, 46 peas, and 17 peppers on it.

Previous action:
take math problem

Here are all the valid actions. Your response should be one of them:
inventory,take 88 eggplants,put math problem in box,take 29 blueberries,take 46 peas,look around,take 22 cabbages,take 17 peppers,take 42 avocados,read math problem,put math problem in showcase

---

## D  Additional Error Analyses

**Why does GPT-4 perform poorly in the SORT­ING game?** In the SORTING game, an agent needs to put 3-5 objects into the answer box in order. In most of its error cases, the GPT-4 agent puts the first one or two objects into the answer box correctly, but it forgets what it did later and takes objects from the answer box out and put them back in a wrong order. This suggests that this GPT-4 agent does not have a good long-term memory of the task.

**Why does the self-supervised model need much more steps to win in the TWC game?** In the TWC games, agents will not fail immediately if it puts an object at a wrong location. While the self-supervised achieved a similar score on the TWC task, it makes more wrong decisions during the process, resulting in a larger number of steps required to complete the task.