# OpenReview forum: "Self-Supervised Behavior Cloned Transformers are Path Crawlers for Text Games"
_EMNLP/2023/Conference — EMNLP 2023 Findings_

### Official Review · Reviewer_TBi2 · 2023-08-03

**Soundness:** 4

**Excitement:**

3: Ambivalent: It has merits (e.g., it reports state-of-the-art results, the idea is nice), but there are key weaknesses (e.g., it describes incremental work), and it can significantly benefit from another round of revision. However, I won't object to accepting it if my co-reviewers champion it.

**Paper Topic And Main Contributions:**

This paper explores the viability of generating self-supervised trajectories to train a decision-transformer style agent for text-based games. The authors present a simple iterative expansion technique to create training data where a path crawler is first employed to explore the game initially for a set of short paths which are then grouped and evaluated by training small T5 style agents on the created path groups and evaluating them against a held out set. The best-performing path groups are then expanded upon iteratively until a desired target is reached. Experiments comparing the proposed data creation technique on 3 separate games show that this self-supervised method outperforms a DRRN baseline on all games and approaches the score reached by a comparable agent trained on gold trajectories.

**Questions For The Authors:**

1. The proposed method provides a way to create synthetic trajectories quickly. Of course, faster is better but in C.2, the authors say that the entire training process finishes in 2-4 hours. Since the synthetic data creation is offline, why is time the variable to be minimized?


2. In Table 1, why does the self-supervised model take a significantly larger number of steps on TWC? Additionally, including standard error bars on Table 1 would be very informative since the test set is only 100 examples


3. By building upon trajectories that were selected in the previous iteration, the proposed technique seems to sacrifice some notion of diversity. Did you have any experiments or ablations varying group size etc. to see if this effect occurs? This would be exacerbated at larger action spaces.

**Reasons To Accept:**

The method is simple and intuitive, almost like a beam search of trajectories and shows solid empirical results on all three games. The writing is clear and easy to follow.


The proposed technique is also general enough to be applied to more complicated game settings with tweaks.

**Reasons To Reject:**

On concern with Table 1 is that the self-supervised method is only compared to an oracle i.e. the supervised model, and two baselines to which it is not directly comparable since the way these are trained are so different i.e. DRRN or GPT4. It would be significantly improved by including a baseline that ablates the effect of the data creation using the iterative method i.e. a random path crawler that generates an equal number of synthetic paths which is then used to train the agent. This would show the effect of how helpful it is to perform the iterative selection step.

**Reproducibility:**

4: Could mostly reproduce the results, but there may be some variation because of sample variance or minor variations in their interpretation of the protocol or method.

**Reviewer Confidence:**

3: Pretty sure, but there's a chance I missed something. Although I have a good feel for this area in general, I did not carefully check the paper's details, e.g., the math, experimental design, or novelty.

---

> ### Author Rebuttal · Authors · 2023-08-29
>
> ## Response to Reasons to Reject:
> ### Reviewer:
> “On concern with Table 1 is that the self-supervised method is only compared to an oracle i.e. the supervised model, and two baselines to which it is not directly comparable since the way these are trained are so different i.e. DRRN or GPT4. It would be significantly improved by including a baseline that ablates the effect of the data creation using the iterative method i.e. a random path crawler that generates an equal number of synthetic paths which is then used to train the agent. This would show the effect of how helpful it is to perform the iterative selection step.”
>
> ### Response:
> Thank you for your suggestion. The primary narrative is that we have developed a self-supervised behavior cloning model that achieves nearly the same performance as a fully-supervised behavior cloning model. The other baselines (DRRN, GPT-4) are provided for context only, and (as you say) use somewhat different solution methods and as such are not directly comparable.  The primary experimental comparison is between supervised vs self-supervised.
>
> With respect to your suggested ablation study (random path crawler), we think that we have already conducted a similar study that we removed from the paper for space.  In that study (which we think of as a noise sensitivity study), we inject a controlled percentage of random paths into each path group, to empirically measure how robust the self-supervision method is to finding paths within noisy groups.  (But note, in this case the paths aren’t completely random: they still arrive at a winning state to the task, they just likely take a solution path that isn’t generalizable).  At the extreme, in this noise analysis we train the T5-base model on paths that are entirely from these random groups.  In these cases, one might hypothesize that the model is being fine tuned on the general protocol of text games in particular (e.g. input: observation, history; output: next action), and may distill some aspects of the task from whatever subparts of the paths are good, but that must ultimately also rely on its internal pre-trained knowledge to do well at the task.  In this case, the model performance decreases substantially (0.21 vs 0.54 on Arithmetic, 0.37 vs 0.72 on Sorting, and 0.68 vs 0.90 on TWC).  This likely speaks to your question, and empirically demonstrates the utility of the proposed method in its impact on performance.  If accepted, we can include this additional analysis (or a similar analysis with true randomly-selected paths, for noise) in the extra page, or in the Appendix, as space allows.
>
> ## Response to Questions for Authors:
> ### Question 1:
> *The proposed method provides a way to create synthetic trajectories quickly. Of course, faster is better but in C.2, the authors say that the entire training process finishes in 2-4 hours. Since the synthetic data creation is offline, why is time the variable to be minimized?*
>
> **Response:** We will clarify this point in the paper. The proposed method is not entirely offline (i.e. the pathcrawler identifies candidate paths, then the generality of those paths is empirically evaluated by training T5 models using the found paths and evaluating their performance on the development set) – so generally you still do want to minimize time. In the worst case, if you had to explore all possible groups of paths, one may have to train an intractibly large number of T5 models. Indeed, one of the major contributions of this paper is the heuristic described in Section 2 (Evaluating Path Groups, and Figure 2) that allows pathfinding to become tractable for self-supervision.
>
> ### Question 2:
> *In Table 1, why does the self-supervised model take a significantly larger number of steps on TWC? Additionally, including standard error bars on Table 1 would be very informative since the test set is only 100 examples*
>
> **Response:** Thank you for this interesting question.  To answer this, we conducted an error analysis on the playthroughs of our agent in the TWC game. It turns out that in the TWC game, agents can try to put objects in the wrong place without any punishments. The self-supervised model usually tries several different places before it finds the correct location, resulting in a larger number of steps to complete.  If accepted, we can include this additional error analysis in the body of the paper or Appendix.
>
> ### Question 3:
> *By building upon trajectories that were selected in the previous iteration, the proposed technique seems to sacrifice some notion of diversity. Did you have any experiments or ablations varying group size etc. to see if this effect occurs? This would be exacerbated at larger action spaces.*
>
> **Response:** This is a valid criticism of both this reinforcement learning model, but also essentially all reinforcement learning models (which have to balance exploration versus exploitation). One could find or build problems where an incorrect action at the start of a game causes failure many steps later. (Take Zork, a text dungeon adventure game developed in the 1970s as an example. Forgetting a particular game-critical item at the start of Zork that you need later in the game can result in game failure). But, to the best of our knowledge, there are no solutions for this – indeed, the very best models solve only 10-12% of more complex games like Zork. Here, we demonstrate that for a selection of challenging benchmarks, we’re able to get near supervised performance with this self-supervised model as formulated. Follow-on work could certainly address more complex reinforcement learning problems (like long-distance dependencies) and compare this algorithm to others. With respect to varying hyperparameters (like group size, etc.), we did vary these hyperparameters during development, but they appear to only affect convergence time, not ultimate performance (except at extremes).  Ultimately, while this work doesn’t solve every problem in reinforcement learning, it does empirically demonstrate that a class of exciting new models (behavior cloning) can be converted from fully-supervised to fully-self-supervised models for many text games that existing agents struggle with.  While that is incremental progress, we feel that it is a relatively large, and potentially impactful step to this subfield that is both exciting and sound.

---

### Official Review · Reviewer_AHeS · 2023-08-04

**Soundness:** 3

**Excitement:**

2: Mediocre: This paper makes marginal contributions (vs non-contemporaneous work), so I would rather not see it in the conference.

**Paper Topic And Main Contributions:**

In this paper, the authors aim to reduce the data dependency of Behavior Cloning Transformers. The authors propose an approach that automatically generates data to train models that solve a class of text games. Specifically, the authors argue that those models can be trained in a self-supervised manner, where the data is generated through crawling, grouping, and evaluating generality. They conduct experiments on three benchmarks: Textworld Common Sense (TWC), Arithmetic, and Sorting. The results show that the proposed method is close to the supervised model while surpassing two other baselines.

The highlights include:
* The text games are interesting benchmarks that mark the capability of language models to solve real-world problems.
* The proposed method is intuitive and easy to understand. This might potentially increase the impact of this work.

The weaknesses are:
* The generalization of this method might be poor as it heavily relies on the forms of games.
* The description of the method is not clear.
* Lack of important baselines.

**Reasons To Accept:**

* The text games are interesting benchmarks that mark the capability of language models solving real-world problems.
* The proposed method is intuitive and easy to understand. This might potentially increase the impact of this work.

**Reasons To Reject:**

* The generalization of this method might be poor as it heavily relies on the forms of games. Specifically, the authors set the grouping heuristic as the actions in each text game. However, this practice is not generalizable as it is not likely to expect all text games to fit this heuristic. Therefore, the proposed method might be limited to a small set of tasks.
* The description of the method is not clear or formal. In Section 2, the authors illustrate their method mainly by figures and examples. Although it might suffice to describe their implementation, it is unclear how readers should adapt this method to their test cases in a principled way.
* Lack of important baselines. The GPT-4 baseline is only evaluated in a zero-shot fashion. However, in recent years, the community has developed multiple ways to properly prompt the model. The authors should evaluate these recent techniques to make sure the baseline is proper.

**Reproducibility:**

4: Could mostly reproduce the results, but there may be some variation because of sample variance or minor variations in their interpretation of the protocol or method.

**Reviewer Confidence:**

3: Pretty sure, but there's a chance I missed something. Although I have a good feel for this area in general, I did not carefully check the paper's details, e.g., the math, experimental design, or novelty.

---

> ### Author Rebuttal · Authors · 2023-08-29
>
> ## Response to Reasons to Reject:
>
> ### Reviewer:
> “The generalization of this method might be poor as it heavily relies on the forms of games. Specifically, the authors set the grouping heuristic as the actions in each text game. However, this practice is not generalizable as it is not likely to expect all text games to fit this heuristic. Therefore, the proposed method might be limited to a small set of tasks.”
>
> **Response:**
> This is incorrect. The only assumption we have about the form of the text games is that each text game has a set of parametric variations, which is a common setting in most recent text game benchmarks (e.g. “TextWorld: A Learning Environment for Text-based Games”, Cote 2018; “Text-based RL Agents with Commonsense Knowledge: New Challenges, Environments and Baselines”, Murugesan et al. 2021, “ScienceWorld: Is your Agent Smarter than a 5th Grader?”, Wang et al. 2022, “TextWorldExpress”, Jansen et al. 2023). Otherwise, all games are formulated as the standard formulation of reinforcement learning problems for text games (e.g. a partially-observable Markov decision process (POMPD) with a known action space and that return an observation and a reward after each action).
>
> While it is a valid criticism that the proposed self-supervised model may not perform well if the action space is huge and the reward is sparse, this is true of /all/ known reinforcement learning methods. We feel that the excitement score is not indicative of the impact of this work – i.e. the performance of a fully-supervised behavior cloned model purely through self-supervision.
>
> ### Reviewer:
> “The description of the method is not clear or formal. In Section 2, the authors illustrate their method mainly by figures and examples. Although it might suffice to describe their implementation, it is unclear how readers should adapt this method to their test cases in a principled way.”
>
> **Response:** Space is limited in the main body of the paper, if accepted, we will use the extra page to include additional implementation details including an algorithm listing.  Adapting this method to other test cases is not difficult as the code of this work is released as open source and our trained models can be used as-is or adapted by others.
>
> ### Reviewer:
> Lack of important baselines. The GPT-4 baseline is only evaluated in a zero-shot fashion. However, in recent years, the community has developed multiple ways to properly prompt the model. The authors should evaluate these recent techniques to make sure the baseline is proper.
>
> **Response:** We (respectfully) strongly push back on this comment.  The purpose of this paper is to show a novel method that archives near-supervised performance from a self-supervised behavior cloning model.  GPT-4 performance (and indeed any other model performance) is unrelated to this, and provided only for context.
> Further, even though the GPT-4 model is orders of magnitude larger and trained on vastly more data than the T5-base model, it still finds some of our benchmarks challenging, indicating that our benchmarks are non-trivial to solve.
> Based on the review policies (https://2023.aclweb.org/blog/review-acl23/, Section 2, “The authors could also do [extra experiment X]”), performing a study on different prompting strategies for GPT-4 would appear tangential to the present study on self-supervision, and is not an appropriate reason for rejection.  We have offered strong empirical evidence to support the primary claims of the paper (supervised performance in a self-supervised behavior cloning model), and the benchmarks are standard benchmarks from TextWorldExpress.  Building a better GPT-4 agent is beyond the scope of this paper, and, even if the GPT-4 agent achieved perfect performance, would not detract from the narrative of the paper – that a self-supervised behavior cloning method can achieve supervised performance.  We believe the current soundness (2) and excitement (2) scores do not reflect the empirical quality of this work, or the potential impact – a small, self-supervised model that achieves near fully-supervised performance is a truly remarkable discovery, even if scoped exclusively to the field of text games.

---

### Official Review · Reviewer_U234 · 2023-08-09

**Soundness:** 3

**Excitement:**

2: Mediocre: This paper makes marginal contributions (vs non-contemporaneous work), so I would rather not see it in the conference.

**Paper Topic And Main Contributions:**

- This paper proposes a behavior cloning-based approach for text games which simulates supervised approaches without the need for ground truth demonstrations. The approach involves (1) generating a tree of possible action sequences in the text game, (2) merging related sequences into groups based on macro-action predicates, and (3) training small language models on these merged groups and evaluating which models generalize to unseen examples in the text game.
- This paper evaluates its method on three benchmarks: (a) Arithmetic (which involves a simple math problem and a pick-and-place task); (b) Sorting (which requires agents to rearrange objects by quantity and reason over different units of measurement); and (c) the TextWorld Common Sense task.
- The proposed model outperforms a DRRN baseline on all three tasks; it does not perform as well as GPT-4 on the arithmetic and commonsense tasks, but outperforms GPT-4 on the sorting task. The model approaches supervised approaches but does not quite close the gap between supervised and self-supervised models for these domains.

**Questions For The Authors:**

Based on my understanding, the sorting task is extremely simple, and it doesn’t rely on any information about the item types. Do you have a guess as to why GPT-4 performs so poorly at this task? Based on the results in Table 1, it seems to be performing roughly at chance, which is surprising to me. Could you include the prompts used in future versions of this paper?

**Reasons To Accept:**

The paper is well-written and the method clear. Despite not performing as well as GPT-4 on some tasks, the T5-based model is substantially smaller and approaches the performance ceiling defined by supervised behavior cloning approaches. To the best of my knowledge, the paper uses reasonable baselines and provides a clear description of tradeoffs between models.

**Reasons To Reject:**

The paper’s primary contribution seems unlikely to generalize beyond text games: in particular, searching over all possible actions (up to the initial reward) is only possible because language in text games is highly restricted (and we assume access to the set of valid actions). Additionally, path grouping based on generalization seems to work only because the tasks evaluated in this paper are limited in their complexity, such that many instances of a task can be solved with the same sequence of macro-level actions. Furthermore, intermediate rewards will not exist in most real-world tasks, limiting the usefulness of incremental path crawling.

**In response to the rebuttal**: I do not believe that these criticisms violate the *CL reviewing guidelines as the authors state. The motivation behind text games is to "simulate complex natural language problems in controllable settings" (Osborne, et al. 2022, cited in the author response). As a result, it is not at all unreasonable to suggest that methods for text games should (at least attempt to) generalize to the real-world domains which they are intended to simulate. After reading the author response, I have chosen to keep my score unchanged.

**Reproducibility:**

4: Could mostly reproduce the results, but there may be some variation because of sample variance or minor variations in their interpretation of the protocol or method.

**Reviewer Confidence:**

2: Willing to defend my evaluation, but it is fairly likely that I missed some details, didn't understand some central points, or can't be sure about the novelty of the work.

---

> ### Author Rebuttal · Authors · 2023-08-29
>
> ## Response to Reasons to Reject:
>
> ### Reviewer:
>
> “The paper’s primary contribution seems unlikely to generalize beyond text games: in particular, searching over all possible actions (up to the initial reward) is only possible because language in text games is highly restricted (and we assume access to the set of valid actions). Additionally, path grouping based on generalization seems to work only because the tasks evaluated in this paper are limited in their complexity, such that many instances of a task can be solved with the same sequence of macro-level actions. Furthermore, intermediate rewards will not exist in most real-world tasks, limiting the usefulness of incremental path crawling.”
>
> ### Response:
>
> **Text games as niche area:** One of the central criticisms of this review is that the work is only applicable to the niche of text games/interactive environments, which violates the ACL reviewing policy (https://2023.aclweb.org/blog/review-acl23/, Section 2, “Topic is too niche – A main track paper may well make a big contribution to a narrow subfield.”).  While text games and interactive environments are a relatively new subfield to NLP conferences, with papers starting to appear only about 5 years ago, they have rapidly grown into a rich subfield with approximately 200 papers published in only a few years (see “A Survey of Text Games for Reinforcement Learning Informed by Natural Language”, Osborne et al. 2022; “A Systematic Survey of Text Worlds as Embodied Natural Language Environments”, Jansen 2022; for review). We feel that the excitement score is not indicative of the impact of this work to this rapidly growing subfield.
>
> **Negative comments on parametric variations, though these are standard research methods:** The reviewer critiques the use of parametric variations of text games (“many instances of a task can be solved with the same sequence of macro-level actions”), even though this is standard research methods in the subfield.  Earlier on in the subfield, agent evaluation was primarily conducted on stand-alone interactive fiction games (e.g. Zork), or on benchmarks (e.g. “Interactive Fiction Games: A Colossal Adventure”, Hausknecht et al.) that combine a large number of interactive fiction games split into train, development, and test sets. However, text games and their respective action spaces can be so different that agents can hardly generalize from one game to another, and task performance was (and still is, to a degree) very low. **Recent text game benchmarks** (e.g. “TextWorld: A Learning Environment for Text-based Games”, Cote 2018; “Text-based RL Agents with Commonsense Knowledge: New Challenges, Environments and Baselines”, Murugesan et al. 2021, “ScienceWorld: Is your Agent Smarter than a 5th Grader?”, Wang et al. 2022, “TextWorldExpress”, Jansen et al. 2023) **all control for this difficulty by using the idea of parametric variations of a given task.**  With parametric variations, a simulator may generate 300 variations of a task (split between train, development, and test sets), with each variation varying both task-critical objects (e.g. the item of clothing that has to be picked up in TWC) as well as environment details (e.g. the environment might have a kitchen and pantry in one variation, and a garage and driveway in another).
>
> **Unlikely to generalize to sparse-reward conditions:** While it is true that this method would likely perform poorly on long-horizon sparse-reward tasks, that is not a criticism of this method, but rather the entire field of reinforcement learning.  Indeed, there is no known reinforcement learning method that functions in such cases.  That being said, we note that our method has been empirically shown to generalized to game variations with different number of intermediate rewards (in the the sorting game, each variation has 3-5 objects to sorting) and different solving paths (in the TWC game, some variations need to open a container first while the others do not).  Classical strong deep reinforcement learning models (e.g. DRRN) – that, shockingly, still outperform most fancy unsupervised (e.g. non-behavior cloning) models for text games, still substantially underperform the method proposed in this work.
>
> ## Response to Questions for Authors:
>
> **Question A:** Based on my understanding, the sorting task is extremely simple, and it doesn’t rely on any information about the item types. Do you have a guess as to why GPT-4 performs so poorly at this task? Based on the results in Table 1, it seems to be performing roughly at chance, which is surprising to me. Could you include the prompts used in future versions of this paper?
> Response: This is an interesting question to answer. We conducted an error analysis on the playthroughs of the GPT-4 to discover the answer.
>
> In the sorting game, the agent must sort 3-5 objects of random quantity in the environment (e.g. 5 apples, 10 pears, 7 oranges) by picking them up, then placing them in an “answer box”. A correct action sequence might be “take 5 apples”, “place 5 apples in answer box”, “take 7 oranges”, “put 7 oranges in answer box”, etc. In our error analysis, GPT4 generally appears to correctly pick the first one or two items and place them in the answer box correctly.  But after that, it frequently tries to take items back out of the answer box, rather than put the next-ordered item in the box. This suggests that it has challenges in keeping track of its task progress. The Sorting game in the benchmark we use (TextWorldExpress) has a draconian scoring function where making an error typically results in a very low or zero score – so if an agent can’t solve the full task, the scoring function prefers it to make partial high-confidence progress rather than make guesses.

---

### Meta-Review · Area_Chair_3avo · 2023-09-19

**Recommendation:** 3

**Metareview:**

The authors proposes a behavior cloning-based approach for text games which simulates supervised approaches without the need for ground truth demonstrations.
The 2 out of 3 reviewers selected "Good" for soundness (1 selected "Strong")

---

### Decision · Program_Chairs · 2023-10-07

**Decision:**

Accept-Findings

**Comment:**

The authors proposes a behavior cloning-based approach for text games which simulates supervised approaches without the need for ground truth demonstrations.
The 2 out of 3 reviewers selected "Good" for soundness (1 selected "Strong")